# Multiscale Semi-Markov Dynamics for Intracortical Brain-Computer Interfaces

**Daniel J. Milstein** [*]
daniel_milstein@alumni.brown.edu

**Jason L. Pacheco** [†]
pachecoj@mit.edu

**Leigh R. Hochberg** [‡ § ¶]
leigh_hochberg@brown.edu

**John D. Simeral** [‡ §]
john_simeral@brown.edu

**Beata Jarosiewicz** [‖ § **]
beataj@stanford.edu

**Erik B. Sudderth** [†† *]
sudderth@uci.edu

## Abstract

Intracortical brain-computer interfaces (iBCIs) have allowed people with tetraplegia to control a computer cursor by imagining the movement of their paralyzed arm or hand. State-of-the-art decoders deployed in human iBCIs are derived from a Kalman filter that assumes Markov dynamics on the angle of intended movement, and a unimodal dependence on intended angle for each channel of neural activity. Due to errors made in the decoding of noisy neural data, as a user attempts to move the cursor to a goal, the angle between cursor and goal positions may change rapidly. We propose a dynamic Bayesian network that includes the on-screen goal position as part of its latent state, and thus allows the person's intended angle of movement to be aggregated over a much longer history of neural activity. This multiscale model explicitly captures the relationship between instantaneous angles of motion and long-term goals, and incorporates semi-Markov dynamics for motion trajectories. We also introduce a multimodal likelihood model for recordings of neural populations which can be rapidly calibrated for clinical applications. In offline experiments with recorded neural data, we demonstrate significantly improved prediction of motion directions compared to the Kalman filter. We derive an efficient online inference algorithm, enabling a clinical trial participant with tetraplegia to control a computer cursor with neural activity in real time. The observed kinematics of cursor movement are objectively straighter and smoother than prior iBCI decoding models without loss of responsiveness.

## 1 Introduction

Paralysis of all four limbs from injury or disease, or *tetraplegia*, can severely limit function, independence, and even sometimes communication. Despite its inability to effect movement in muscles, neural activity in motor cortex still modulates according to people's intentions to move their paralyzed arm or hand, even years after injury [Hochberg et al., 2006, Simeral et al., 2011, Hochberg et al.,

[*]Department of Computer Science, Brown University, Providence, RI, USA.

[†]Computer Science and Artificial Intelligence Laboratory, MIT, Cambridge, MA, USA.

[‡]School of Engineering, Brown University, Providence, RI, USA; and Department of Neurology, Massachusetts General Hospital, Boston, MA, USA.

[§]Rehabilitation R&D Service, Department of Veterans Affairs Medical Center, Providence, RI, USA; and Brown Institute for Brain Science, Brown University, Providence, RI, USA.

[¶]Department of Neurology, Harvard Medical School, Boston, MA, USA.

[‖]Department of Neuroscience, Brown University, Providence, RI, USA.

[**]Present affiliation: Dept. of Neurosurgery, Stanford University, Stanford, CA, USA.

[††]Department of Computer Science, University of California, Irvine, CA, USA.

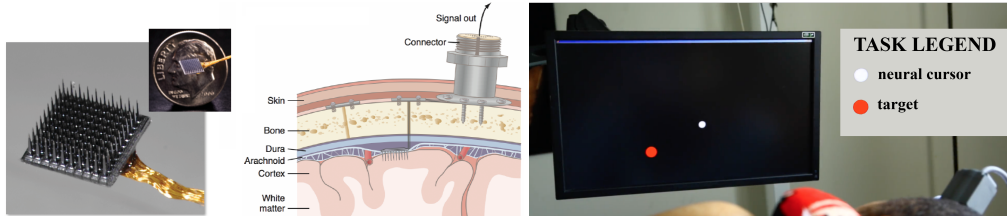

Figure 1: A microelectrode array (*left*) is implanted in the motor cortex (*center*) to record electrical activity. Via this activity, a clinical trial participant (*right*, lying on his side in bed) then controls a computer cursor with an iBCI. A cable connected to the electrode array via a transcutaneous connector (gray box) sends neural signals to the computer for decoding. Center drawing from Donoghue et al. [2011] and used with permission of the author. The right image is a screenshot of a video included in the supplemental material that demonstrates real time decoding via our MSSM model.

2012, Collinger et al., 2013]. *Intracortical brain-computer interfaces* (iBCIs) utilize neural signals recorded from implanted electrode arrays to extract information about movement intentions. They have enabled individuals with tetraplegia to control a computer cursor to engage in tasks such as on-screen typing [Bacher et al., 2015, Jarosiewicz et al., 2015, Pandarinath et al., 2017], and to regain volitional control of their own limbs [Ajiboye et al., 2017].

Current iBCIs are based on a Kalman filter that assumes the vector of desired cursor movement evolves according to Gaussian random walk dynamics, and that neural activity is a Gaussian-corrupted linear function of this state [Kim et al., 2008]. In Sec. 2, we review how the Kalman filter is applied to neural decoding, and studies of the motor cortex by Georgopoulos et al. [1982] that justify its use. In Sec. 3, we improve upon the Kalman filter's linear observation model by introducing a flexible, multimodal likelihood inspired by more recent research [Amirikian and Georgopulos, 2000]. Sec. 4 then proposes a graphical model (a dynamic Bayesian network [Murphy, 2002]) for the relationship between the angle of intended movement and the intended on-screen goal position. We derive an efficient inference algorithm via an online variant of the junction tree algorithm [Boyen and Koller, 1998]. In Sec. 5, we use recorded neural data to validate the components of our *multiscale semi-Markov* (MSSM) model, and demonstrate significantly improved prediction of motion directions in offline analysis. Via a real time implementation of the inference algorithm on a constrained embedded system, we then evaluate online decoding performance as a participant in the BrainGate2[1] iBCI pilot clinical trial uses the MSSM model to control a computer cursor with his neural activity.

## 2 Neural decoding via a Kalman filter

The Kalman filter is the current state-of-the-art for iBCI decoding. There are several configurations of the Kalman filter used to enable cursor control in contemporary iBCI systems [Pandarinath et al., 2017, Jarosiewicz et al., 2015, Gilja et al., 2015] and there is no broad consensus in the iBCI field on which is most suited for clinical use. In this paper, we focus on the variant described by Jarosiewicz et al. [2015].

Participants in the BrainGate2 clinical trial receive one or two microelectrode array implants in the motor cortex (see Fig. 1). The electrical signals recorded by this electrode array are then transformed (via signal processing methods designed to reduce noise) into a $D$-dimensional neural activity vector $z_t \in \mathbb{R}^D$, sampled at 50 Hz. From the sequence of neural activity, the Kalman filter estimates the latent state $x_t \in \mathbb{R}^2$, a vector pointing in the intended direction of cursor motion. The Kalman filter assumes a jointly Gaussian model for cursor dynamics and neural activity,

$$x_t \mid x_{t-1} \sim \mathcal{N}(Ax_{t-1}, W), \qquad z_t \mid x_t \sim \mathcal{N}(b + Hx_t, Q), \tag{1}$$

with cursor dynamics $A \in \mathbb{R}^{2 \times 2}$, process noise covariance $W \in \mathbb{R}^{2 \times 2}$, and (typically non-diagonal) observation covariance $Q \in \mathbb{R}^{D \times D}$. At each time step, the on-screen cursor's position is moved by the estimated latent state vector (decoder output) scaled by a constant, the speed gain.

The function relating neural activity to some measurable quantity of interest is called a *tuning curve*. A common model of neural activity in the motor cortex assumes that each neuron's activity is highest

for some preferred direction of motion, and lowest in the opposite direction, with intermediate activity often resembling a cosine function. This *cosine tuning model* is based on pioneering studies of the motor cortex of non-human primates [Georgopoulos et al., 1982], and is commonly used (or implicitly assumed) in iBCI systems because of its mathematical simplicity and tractability.

Expressing the inner product between vectors via the cosine of the angle between them, the expected neural activity of the $j^{\text{th}}$ component of Eq. (1) can be written as

$$\mathbb{E}[z_{tj} \mid x_t] = b_j + h_j^T x_t = b_j + ||x_t|| \cdot ||h_j|| \cdot \cos\left(\theta_t - \text{atan}\left(\frac{h_{j2}}{h_{j1}}\right)\right), \qquad (2)$$

where $\theta_t$ is the intended angle of movement at timestep $t$, $b_j$ is the baseline activity rate for channel $j$, and $h_j$ is the $j^{\text{th}}$ row of the observation matrix $H = (h_1^T, \ldots, h_D^T)^T$. If $x_t$ is further assumed to be a unit vector (a constraint *not* enforced by the Kalman filter), Eq. (2) simplifies to $h_j^T x_t = m_j \cos(\theta_t - p_j)$, where $m_j$ is the modulation of the tuning curve and $p_j$ specifies the angular location of the peak of the cosine tuning curve (the preferred direction). Thus, cosine tuning models are linear.

To collect labeled training data for decoder calibration, the participant is asked to attempt to move a cursor to prompted target locations. We emphasize that although the clinical random target task displays only one target at a time, this target position is unknown to the decoder. Labels are constructed for the neural activity patterns by assuming that at each 20ms time step, the participant intends to move the cursor straight to the target [Jarosiewicz et al., 2015, Gilja et al., 2015]. These labeled data are used to fit the observation matrix $H$ and neuron baseline rates (biases) $b$ via ridge regression. The observation noise covariance $Q$ is estimated as the empirical covariance of the residuals. The state dynamics matrix $A$ and process covariance matrix $W$ may be tuned to adjust the responsiveness of the iBCI system.

## 3 Flexible tuning likelihoods

The cosine tuning model reviewed in the previous section has several shortcomings. First, motor cortical neurons that have unimodal tuning curves often have narrower peaks that are better described by von Mises distributions [Amirikian and Georgopulos, 2000]. Second, tuning can be multimodal. Third, neural features used for iBCI decoding may capture the pooled activity of several neurons, not just one [Fraser et al., 2009]. While bimodal von Mises models were introduced by Amirikian and Georgopulos [2000], up to now iBCI decoders based on von Mises tuning curves have only employed unimodal mean functions proportional to a single von Mises density [Koyama et al., 2010]. In contrast, we introduce a multimodal likelihood proportional to an arbitrary number of regularly spaced von Mises densities and incorporate this likelihood into an iBCI decoder. Moreover, we can efficiently fit parameters of this new likelihood via ridge regression. Computational efficiency is crucial to allow rapid calibration in clinical applications.

Let $\theta_t \in [0, 2\pi)$ denote the intended angle of cursor movement at time $t$. The *flexible tuning* likelihood captures more complex neural activity distributions via a regression model with nonlinear features:

$$z_t \mid \theta_t \sim \mathcal{N}\left(b + w^T \phi(\theta_t), Q\right), \qquad \phi_k(\theta_t) = \exp\left[\epsilon \cos\left(\theta_t - \varphi_k\right)\right]. \qquad (3)$$

The features are a set of $K$ von Mises basis functions $\phi(\theta) = (\phi_1(\theta), \ldots, \phi_K(\theta))^T$. Basis functions $\phi_k(x)$ are centered on a regular grid of angles $\varphi_k$, and have tunable concentration $\epsilon$.

Using human neural data recorded during cued target tasks, we compare regression fits for the flexible tuning model to the standard cosine tuning model (Fig. 2). In addition to providing better fits for channels with complex or multimodal activity, the flexible tuning model also provides good fits to apparently cosine-tuned signals. This leads to higher predictive likelihoods for held-out data, and as we demonstrate in Sec. 5, more accurate neural decoding algorithms.

## 4 Multiscale Semi-Markov Dynamical Models

The key observation underlying our multiscale dynamical model is that the sampling rate used for neural decoding (typically around 50 Hz) is much faster than the rate that the goal position changes (under normal conditions, every few seconds). In addition, frequent but small adjustments of cursor aim angle are required to maintain a steady heading. State-of-the-art Kalman filter approaches to iBCIs

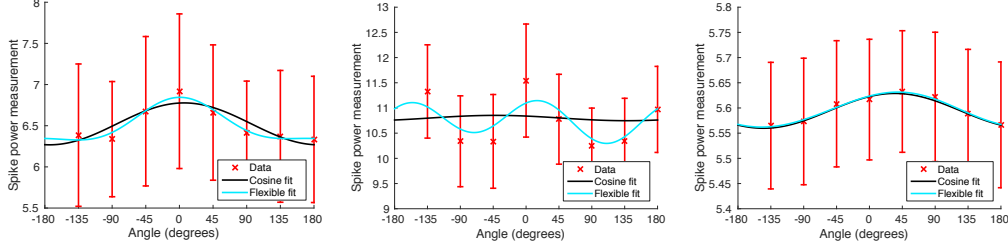

Figure 2: **Flexible tuning curves.** Each panel shows the empirical mean and standard deviation (red) of example neural signals recorded from a single intracortical electrode while a participant is moving within 45 degrees of a given direction in a cued target task. These signals can violate the assumptions of a cosine tuning model (black), as evident in the left two examples. The flexible regression likelihood (cyan) captures neural activity with varying concentration (left) and multiple tuning directions (center), as well as cosine-tuned signals (right). Because neural activity from individual electrodes is very noisy (the standard deviation within each angular bin exceeds the change in mean activity across angles), information from multiple electrodes is aggregated over time for effective decoding.

are incapable of capturing these multiscale dynamics since they assume first-order Markov dependence across time and do not explicitly represent goal position. To cope with this, hyperparameters of the linear Gaussian dynamics must be tuned to simultaneously remain sensitive to frequent directional adjustments, but not so sensitive that cursor dynamics are dominated by transient neural activity.

Our proposed MSSM decoder, by contrast, explicitly represents goal position in addition to cursor aim angle. Through the use of semi-Markov dynamics, the MSSM enables goal position to evolve at a different rate than cursor angle while allowing for a high rate of neural data acquisition. In this way, the MSSM can integrate across different timescales to more robustly infer the (unknown) goal position and the (also unknown) cursor aim. We introduce the model in Sec. 4.1 and 4.2. We derive an efficient decoding algorithm, based on an online variant of the junction tree algorithm, in Sec. 4.3.

## 4.1 Modeling Goals and Motion via a Dynamic Bayesian Network

The MSSM directed graphical model (Fig. 3) uses a structured latent state representation, sometimes referred to as a *dynamic Bayesian network* [Murphy, 2002]. This factorization allows us to discretize latent state variables, and thereby support non-Gaussian dynamics and data likelihoods. At each time $t$ we represent discrete cursor aim $\theta_t$ as 72 values in $[0, 2\pi)$ and goal position $g_t$ as a regular grid of $40 \times 40 = 1600$ locations (see Fig. 4). Each cell of the grid is small compared to elements of a graphical interface. Cursor aim dynamics are conditioned on goal position and evolve according to a smoothed von Mises distribution:

$$\text{vMS}(\theta_t \mid g_t, p_t) \triangleq \alpha/2\pi + (1 - \alpha)\text{vonMises}(\theta_t \mid a(g_t, p_t), \bar{\kappa}). \qquad (4)$$

Here, $a(g, p) = \tan^{-1}((g_y - p_y)/(g_x - p_x))$ is the angle from the cursor $p = (p_x, p_y)$ to the goal $g = (g_x, g_y)$, and the concentration parameter $\bar{\kappa}$ encodes the expected accuracy of user aim. Neural activity from some participants has short bursts of noise during which the learned angle likelihood is inaccurate; the outlier weight $0 < \alpha < 1$ adds robustness to these noise bursts.

## 4.2 Multiscale Semi-Markov Dynamics

The first-order Markov assumption made by existing iBCI decoders (see Eq. (1)) imposes a geometric decay in state correlation over time. For example, consider a scalar Gaussian state-space model: $x_t = \beta x_{t-1} + v, v \sim \mathcal{N}(0, \sigma^2)$. For time lag $k > 0$, the covariance between two states $\text{cov}(x_t, x_{t+k})$ decays as $\beta^{-k}$. This weak temporal dependence is highly problematic in the iBCI setting due to the mismatch between downsampled sensor acquisition rates used for decoding (typically around 50Hz, or 20ms per timestep) and the time scale at which the desired goal position changes (seconds).

We relax the first-order Markov assumption via a semi-Markov model of state dynamics [Yu, 2010]. Semi-Markov models, introduced by Levy [1954] and Smith [1955], divide the state evolution into contiguous segments. A segment is a contiguous series of timesteps during which a latent variable is unchanged. The conditional distribution over the state at time $x_t$ depends not only on the previous state $x_{t-1}$, but also on a duration $d_t$ which encodes how long the state is to remain unchanged:

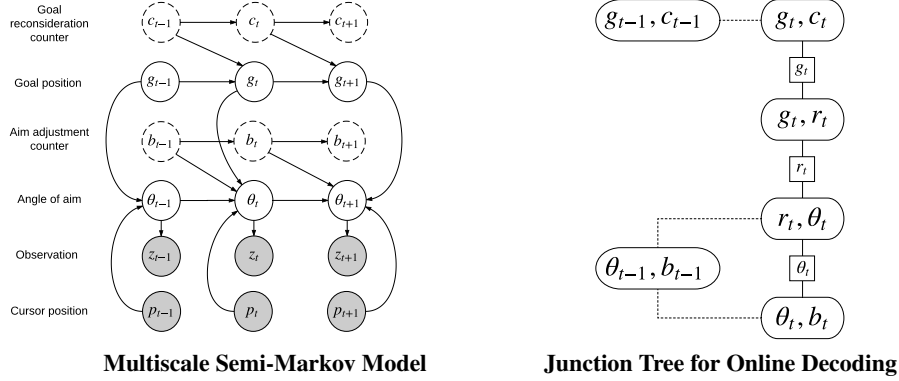

**Multiscale Semi-Markov Model** · **Junction Tree for Online Decoding**

Figure 3: **Multiscale semi-Markov dynamical model.** *Left:* The multiscale directed graphical model of how goal positions $g_t$, angles of aim $\theta_t$, and observed cursor positions $p_t$ evolve over three time steps. Dashed nodes are counter variables enabling semi-Markov dynamics. *Right:* Illustration of the junction tree used to compute marginals for online decoding, as in Boyen and Koller [1998]. Dashed edges indicate cliques whose potentials depend on the marginal approximations at time $t-1$. The inference uses an auxiliary variable $r_t \triangleq a(g_t, p_t)$, the angle from the cursor to the current goal, to reduce computation and allow inference to operate in real time.

$p(x_t \mid x_{t-1}, d_t)$. Duration is modeled via a latent *counter* variable, which is drawn at the start of each segment and decremented deterministically until it reaches zero, at which point it is resampled. In this way the semi-Markov model is capable of integrating information over longer time horizons, and thus less susceptible to intermittent bursts of sensor noise.

We define separate semi-Markov dynamical models for the goal position and the angle of intended movement. As detailed in the supplement, in experiments our duration distributions were uniform, with parameters informed by knowledge about typical trajectory durations and reaction times.

**Goal Dynamics** A counter $c_t$ encodes the temporal evolution of the semi-Markov dynamics on goal positions: $c_t$ is drawn from a discrete distribution $p(c)$ at the start of each trajectory, and then decremented deterministically until it reaches zero. (During decoding we do not know the value of the counter, and maintain a posterior probability distribution over its value.) The goal position $g_t$ remains unchanged until the goal counter reaches zero, at which point with probability $\eta$ we resample a new goal, and we keep the same goal with the remaining probability $1 - \eta$:

$$p(c_t \mid c_{t-1}) = \begin{cases} 1, & c_t = c_{t-1} - 1, c_{t-1} > 0, & \text{Decrement} \\ p(c_t), & c_{t-1} = 0, & \text{Sample new counter} \\ 0, & & \text{Otherwise} \end{cases} \quad (5)$$

$$p(g_t \mid c_{t-1}, g_{t-1}) = \begin{cases} 1, & c_{t-1} > 0, g_t = g_{t-1}, & \text{Goal position unchanged} \\ \eta\frac{1}{G} + (1-\eta), & c_{t-1} = 0, g_t = g_{t-1}, & \text{Sample same goal position} \\ \eta\frac{1}{G}, & c_{t-1} = 0, g_t \neq g_{t-1}, & \text{Sample new goal position} \\ 0, & & \text{Otherwise} \end{cases} \quad (6)$$

**Cursor Angle Dynamics** We define similar semi-Markov dynamics for the cursor angle via an aim counter $b_t$. Once the counter reaches zero, we sample a new aim counter value from the discrete distribution $p(b)$, and a new cursor aim angle from the smoothed von Mises distribution of Eq. (4):

$$p(b_t \mid b_{t-1}) = \begin{cases} 1, & b_t = b_{t-1} - 1, b_{t-1} > 0, & \text{Decrement} \\ p(b_t), & b_{t-1} = 0, & \text{Sample new counter} \\ 0, & & \text{Otherwise} \end{cases} \quad (7)$$

$$p(\theta_t \mid b_{t-1}, \theta_{t-1}, p_t, g_t) = \begin{cases} \theta_{t-1} & b_{t-1} > 0, & \text{Keep cursor aim} \\ \text{vMS}(\theta_t \mid g_t, p_t) & b_{t-1} = 0, & \text{Sample new cursor aim} \end{cases} \quad (8)$$

### 4.3 Decoding via Approximate Online Inference

Efficient decoding is possible via an approximate variant of the junction tree algorithm [Boyen and Koller, 1998]. We approximate the full posterior at time $t$ via a partially factorized posterior:

$$p(g_t, c_t, \theta_t, b_t \mid z_{1...t}) \approx p(g_t, c_t \mid z_{1...t})p(\theta_t, b_t \mid z_{1...t}). \quad (9)$$

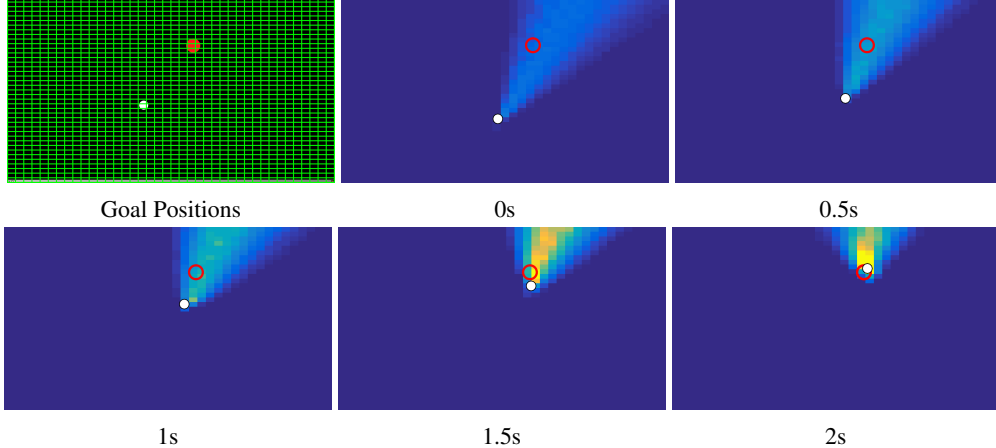

| Goal Positions | 0s | 0.5s |
| 1s | 1.5s | 2s |

Figure 4: **Decoding goal positions.** The MSSM represents goal position via a regular grid of $40 \times 40$ locations (*upper left*). For one real sequence of recorded neural data, the above panels illustrate the motion of the cursor (white dot) to the user's target (red circle). Panels show the marginal posterior distribution over goal positions at 0.5s intervals (25 discrete time steps of graphical model inference). Yellow goal states have highest probability, dark blue goal states have near-zero probability. Note the temporal aggregation of directional cues.

Here $p(g_t, c_t \mid z_{1\ldots t})$ is the marginal on the goal position and goal counter, and $p(\theta_t, b_t \mid z_{1\ldots t})$ is the marginal on the angle of aim and the aim counter. Note that in this setting goal position $g_t$ and cursor aim $\theta_t$, as well as their respective counters $c_t$ and $b_t$, are unknown and must be inferred from neural data. At each inference step we use the junction tree algorithm to compute state marginals at time $t$, conditioned on the factorized posterior approximation from time step $t - 1$ (see Fig. 3). Boyen and Koller [1998] show that this technique has bounded approximation error over time, and Murphy and Weiss [2001] show this as a special case of loopy belief propagation.

Detailed inference equations are derived in the supplemental material. Given $G$ goal positions and $A$ discrete angle states, each temporal update for our online decoder requires $\mathcal{O}(GA + A^2)$ operations. In contrast, the exact junction tree algorithm would require $\mathcal{O}(G^2 A^2)$ operations; for practical numbers of goals $G$, realtime implementation of this exact decoder is infeasible.

Figure 4 shows several snapshots of the marginal posterior over goal position. At each time the MSSM decoder moves the cursor along the vector $\mathbb{E}\left[\frac{g_t - p_t}{\|g_t - p_t\|}\right]$, computed by taking an average of the directions needed to get to each possible goal, weighted by the inferred probability that each goal is the participant's true target. This vector is smaller in magnitude when the decoder is less certain about the direction in which the intended goal lies, which has the practical benefit of allowing the participant to slow down near the goal.

## 5  Experiments

We evaluate all decoders under a variety of conditions and a range of configurations for each decoder. Controlled offline evaluations allow us to assess the impact of each proposed innovation. To analyze the effects of our proposed likelihood and multiscale dynamics in isolation, we construct a baseline *hidden Markov model* (HMM) decoder using the same discrete representation of angles as the MSSM, and either cosine-tuned or flexible likelihoods. Our findings show that the offline decoding performance of the MSSM is superior in all respects to baseline models.

We also evaluate the MSSM decoder in two online clinical research sessions, and compare head-to-head performance with the Kalman filter. Previous studies have tested the Kalman filter under a variety of responsive parameter configurations and found a tradeoff between slow, smooth control versus fast, meandering control [Willett et al., 2016, 2017]. Through comparisons to the Kalman, we demonstrate that the MSSM decoder maintains smoother and more accurate control at comparable speeds. These realtime results are preliminary since we have yet to evaluate the MSSM decoder on other clinical metrics such as communication rate.

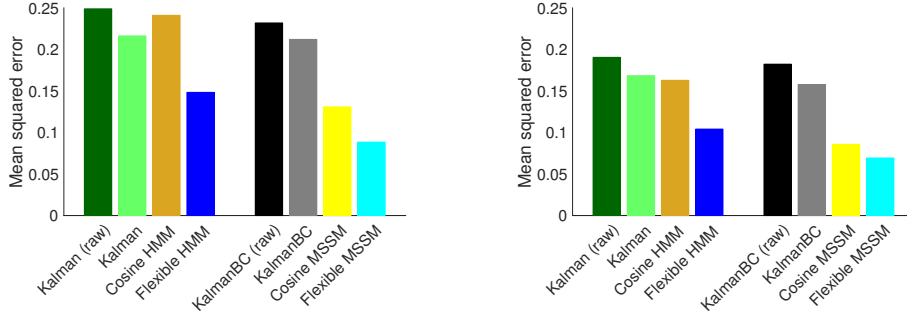

Figure 5: **Offline decoding.** Mean squared error of angular prediction for a variety of decoders, where each decoder processes the same sets of recorded data. We analyze 24 minutes (eight 3-minute blocks) of neural data recorded from participant T9 on trial days 546 and 552. We use one block for testing and the remainder for training, and average errors across the choice of test block. On the left, we report errors over all time points. On the right, we report errors on time points during which the cursor was outside a fixed distance from the target. For both analyses, we exclude the initial 1s after target acquisition, during which the ground truth is unreliable. To isolate preprocessing effects, the plots separately report the Kalman without preprocessing ("raw"). Dynamics effects are isolated by separately evaluating HMM dynamics ("HMM"), and likelihood effects are isolated by separately evaluating flexible likelihood and cosine tuning in each configuration. "KalmanBC" denotes the Kalman filter with an additional kinematic bias-correction heuristic [Jarosiewicz et al., 2015].

## 5.1  Offline evaluation

We perform offline analysis using previously recorded data from two historical sessions of iBCI use with a single participant (T9). During each session the participant is asked to perform a *cued target task* in which a target appears at a random location on the screen and the participant attempts to move the cursor to the target. Once the target is acquired or after a timeout (10 seconds), a new target is presented at a different location. Each session is composed of several 3 minute segments or *blocks*.

To evaluate the effect of each innovation we compare to an HMM decoder. This HMM baseline isolates the effect of our flexible likelihood since, like the Kalman filter, it does not model goal positions and assumes first-order Markov dynamics. Let $\theta_t$ be the latent angle state at time $t$ and $x(\theta) = (\cos(\theta), \sin(\theta))^T$ the corresponding unit vector. We implement a pair of HMM decoders for cosine tuning and our proposed flexible tuning curves,

$$\underbrace{z_t \mid \theta_t \sim \mathcal{N}(b + Hx(\theta_t), Q)}_{\text{Cosine HMM}}, \quad \underbrace{z_t \mid \theta_t \sim \mathcal{N}(b + w^T\phi(\theta_t), Q)}_{\text{Flexible HMM}}$$

Here, $\phi(\cdot)$ are the basis vectors defined in Eq. (3). The state $\theta_t$ is discrete, taking one of 72 angular values equally spaced in $[0, 2\pi)$, the same discretization used by the MSSM. Continuous densities are appropriately normalized. Unlike the linear Gaussian state-space model, the HMMs constrain latent states to be valid angles (equivalently, unit vectors) rather than arbitrary vectors in $\mathbb{R}^2$.

We analyze decoder accuracy within each session using a leave-one-out approach. Specifically, we test the decoder on each held-out block using the remaining blocks in the same session for training. We report MSE of the predicted cursor direction, using the unit vector from the cursor to the target as ground truth, and normalizing decoder output vectors. We used the same recorded data for each decoder. See the supplement for further details.

Figure 5 summarizes the findings of the offline comparisons for a variety of decoder configurations. First, we evaluate the effect of preprocessing the data by taking the square root, applying a low-pass IIR filter, and clipping the data outside a $5\sigma$ threshold, where $\sigma$ is the empirical standard deviation of training data. This preprocessing significantly improves accuracy for all decoders. The MSSM model compares favorably to all configurations of the Kalman decoders. The majority of benefit comes from the semi-Markov dynamical model, but additional gains are observed when including the flexible tuning likelihood. Finally, it has been observed that the Kalman decoder is sensitive to outliers for which Jarosiewicz et al. [2015] propose a correction to avoid biased estimates. We test the Kalman filter with and without this correction.

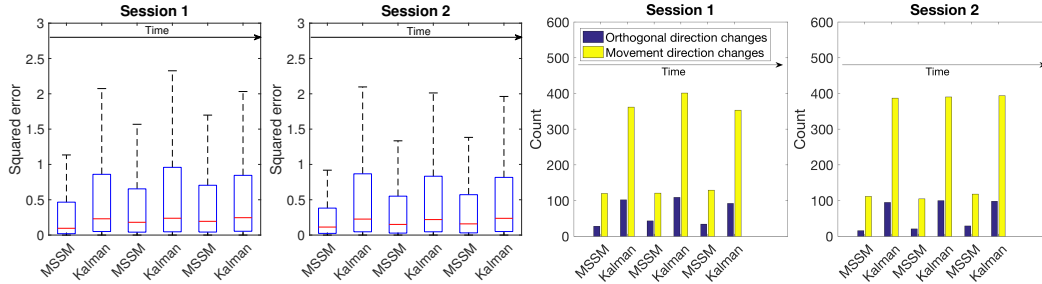

Figure 6: **Realtime decoding.** A realtime comparison of the Kalman filter and MSSM with flexible likelihoods from two sessions with clinical trial participant T10. *Left:* Box plots of squared error between unit vectors from cursor to target and normalized (unit vector) decoder output for each four-minute comparison block in a session. MSSM errors are consistently smaller. *Right:* Two metrics that describe the smoothness of cursor trajectories, introduced by MacKenzie et al. [2001] and commonly used to quantify iBCI performance [Kim et al., 2008, Simeral et al., 2011]. The task axis for a trajectory is the straight line from the cursor's position at the start of a trajectory to a goal. Orthogonal directional changes measure the number of direction changes towards or away from the goal, and movement direction changes measure the number of direction changes towards or away from the task axis. The MSSM decoder shows significantly fewer direction changes according to both metrics.

## 5.2 Realtime evaluation

Next, we examined whether the MSSM method was effective for realtime iBCI control by a clinical trial participant. On two different days, a clinical trial participant (T10) completed six four-minute comparison blocks. In these blocks, we alternated using an MSSM decoder with flexible likelihoods and novel preprocessing, or a standard Kalman decoder. As with the Kalman decoding described in Jarosiewicz et al. [2015], we used the Kalman filter in conjunction with a bias correcting postprocessing heuristic. We used the feature selection method proposed by Malik et al. [2015] to select $D = 60$ channels of neural data, and used these same 60 channels for both decoders.

Jarosiewicz et al. [2015] selected the timesteps of data to use for parameter learning by taking the first two seconds of each trajectory after a 0.3s reaction time. For both decoders, we instead selected all timesteps in which the cursor was a fixed distance from the cued goal because we found this alternative method lead to improvements in offline decoding. Both methods for selecting subsets of the calibration data are designed to compensate for the fact that vectors from cursor to target are not a reliable estimator for participants' intended aim when the cursor is near the target.

**Decoding accuracy.** Figure 6 shows that our MSSM decoder had less directional error than the configuration of the Kalman filter that we compared to. We confirmed the statistical significance of this result using a Wilcoxon rank sum test. To accommodate the Wilcoxon rank sum test's independence assumption, we divided the data into individual trajectories from a starting point towards a goal, that ended either when the cursor reached the goal or at a timeout (10 seconds). We then computed the mean squared error of each trajectory, where the squared error is the squared Euclidean distance between the normalized (unit vector) decoded vectors and the unit vectors from cursor to target. Within each session, we compared the distributions of these mean squared errors for trajectories between decoders ($p < 10^{-6}$ for each session). MSSM also performed better than the Kalman on metrics from MacKenzie et al. [2001] that measure the smoothness of cursor trajectories (see Fig. 6).

Figure 7 shows example trajectories as the cursor moves toward its target via the MSSM decoder or the (bias-corrected) Kalman decoder. Consistent with the quantitative error metrics, the trajectories produced by the MSSM model were smoother and more direct than those of the Kalman filter, especially as the cursor approached the goal. The distance ratio (the ratio of the length of the trajectory to the line from the starting position to the goal) averaged 1.17 for the MSSM decoder and 1.28 for the Kalman decoder, a significant difference (Wilcoxon rank sum test, $p < 10^{-6}$). Some trajectories for both decoders are shown in Figure 7. Videos of cursor movement under both decoding algorithms, and additional experimental details, are included in the supplemental material.

**Decoding speed.** We controlled for speed by configuring both decoders to average the same fast speed determined in collaboration with clinical research engineers familiar with the participant's preferred cursor speed. For each decoder, we collected a block of data in which the participant used

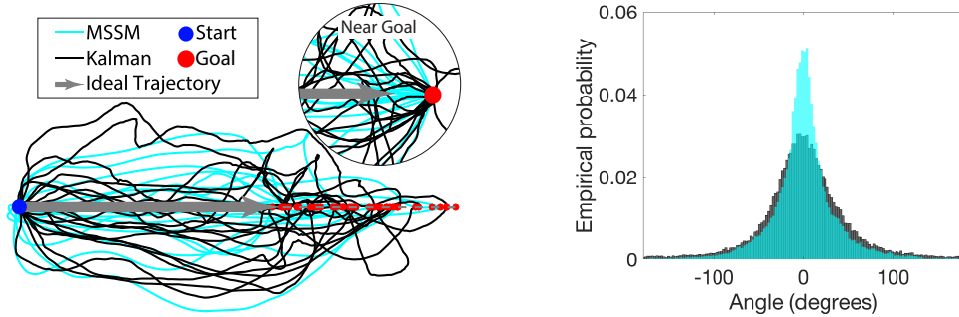

Figure 7: **Examples of realtime decoding trajectories.** *Left*: 20 randomly selected trajectories for the Kalman decoder, and 20 trajectories for the MSSM decoder. The trajectories are aligned so that the starting position is at the origin and rotated so the goal position is on the positive, horizontal axis. The MSSM decoder exhibits fewer abrupt direction changes. *Right:* The empirical probability of instantaneous angle of movement, after rotating all trajectories from the realtime data (24 minutes of iBCI use with each decoder). The MSSM distribution (shown as translucent cyan) is more peaked at zero degrees, corresponding to direct motion towards the goal.

that decoder to control the cursor. For each of these blocks, we computed the trimmed mean of the speed, and then linearly extrapolated the speed gain needed for the desired speed. Although such an extrapolation is approximate, the average times to acquire a target with each decoder at the extrapolated speed gains were within 6% of each other: 2.6s for the Kalman decoder versus 2.7s for the MSSM decoder. This speed discrepancy is dominated by the relative performance improvement of MSSM over Kalman: the Kalman had a 30.7% greater trajectory mean squared error, 249% more orthogonal direction changes, and 224% more movement direction changes.

This approach to evaluating decoder performance differs from that suggested by Willett et al. [2016], which discusses the possibility of optimizing the speed gain and other decoder parameters to minimize target acquisition time. In contrast, we matched the speed of both decoders and evaluated decoding error and smoothness. We did not extensively tune the dynamics parameters for either decoder, instead relying on the Kalman parameters in everyday use by T10. For MSSM we tried two values of $\eta$, which controls the sampling of goal states (6), and chose the remaining parameters offline.

## 6 Conclusion

We introduce a flexible likelihood model and multiscale semi-Markov (MSSM) dynamics for cursor control in intracortical brain-computer interfaces. The flexible tuning likelihood model extends the cosine tuning model to allow for multimodal tuning curves and narrower peaks. The MSSM dynamic Bayesian network explicitly models the relationship between the goal position, the cursor position, and the angle of intended movement. Because the goal position changes much less frequently than the angle of intended movement, a decoder's past knowledge of the goal position stays relevant for longer, and the MSSM model can use longer histories of neural activity to infer the direction of desired movement.

To create a realtime decoder, we derive an online variant of the junction tree algorithm with provable accuracy guarantees. We demonstrate a significant improvement over the Kalman filter in offline experiments with neural recordings, and demonstrate promising preliminary results in clinical trial tests. As seen in the videos, the MSSM decoder yields an appreciably straighter and smoother trajectory than the Kalman decoder. Future work will further evaluate the suitability of this method for clinical use. We hope that the MSSM graphical model will also enable further advances in iBCI decoding, for example by encoding the structure of a known user interface in the set of latent goals.

**Author contributions**   DJM, JLP, and EBS created the flexible tuning likelihood and the multiscale semi-Markov dynamics. DJM derived the inference (decoder), wrote software implementations of these methods, and performed data analyses. DJM, JLP, and EBS designed offline experiments. DJM, BJ, and JDS designed clinical research sessions. LRH is the sponsor-investigator of the BrainGate2 pilot clinical trial. DJM, JLP, and EBS wrote the manuscript with input from all authors.

## Acknowledgments

The authors thank Participants T9 and T10 and their families, Brian Franco, Tommy Hosman, Jessica Kelemen, Dave Rosler, Jad Saab, and Beth Travers for their contributions to this research. Support for this study was provided by the Office of Research and Development, Rehabilitation R&D Service, Department of Veterans Affairs (B4853C, B6453R, and N9228C), the National Institute on Deafness and Other Communication Disorders of National Institutes of Health (NIDCD-NIH: R01DC009899), MGH-Deane Institute, and The Executive Committee on Research (ECOR) of Massachusetts General Hospital. The content is solely the responsibility of the authors and does not necessarily represent the official views of the National Institutes of Health, or the Department of Veterans Affairs or the United States Government. CAUTION: Investigational Device. Limited by Federal Law to Investigational Use.

*Disclosure:* Dr. Hochberg has a financial interest in Synchron Med, Inc., a company developing a minimally invasive implantable brain device that could help paralyzed patients achieve direct brain control of assistive technologies. Dr. Hochberg's interests were reviewed and are managed by Massachusetts General Hospital, Partners HealthCare, and Brown University in accordance with their conflict of interest policies.

## Footnotes

[1]Caution: Investigational Device. Limited by Federal Law to Investigational Use.

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
