[Supplementary Material · supplementalMaterial.pdf]

# Supplemental material for
# Multiscale Semi-Markov Dynamics
# for Intracortical Brain-Computer Interfaces

**Daniel J. Milstein** [*]
daniel_milstein@alumni.brown.edu

**Jason L. Pacheco** [†]
pachecoj@mit.edu

**Leigh R. Hochberg** [‡ § ¶]
leigh_hochberg@brown.edu

**John D. Simeral** [‡ §]
john_simeral@brown.edu

**Beata Jarosiewicz** [∥ § **]
beataj@stanford.edu

**Erik B. Sudderth** [†† *]
sudderth@uci.edu

## 1  Offline testing details

Our signal processing pipeline extracts two types of information from each electrode at 50Hz: a continuous signal (spike power) that gives the amount of power in the spike frequency range (Homer et al., 2013) and a discrete spike count (Masse et al., 2014). The models we present focus on continuous emissions and only use the spike power signal. In our offline comparisons, we did not use feature selection for these comparisons, instead using all spike power channels available.

For both offline and real-time testing, we used an additional preprocessing step for the calibration data, coincident noise removal (Jarosiewicz et al., 2015), to compensate for how the neural recordings are subject to extreme outliers across many channels at once. At each timestep, we summed the observations for all channels together, and computed the mean and standard deviation of the middle 95% of these values. When the sum for a timestep was more than 10 of these trimmed standard deviations from the trimmed mean, all observations at that timestep were set to the channel mean.

We evaluated our methods offline in two historical sessions. Each session had four blocks of unassisted cursor control, each three minutes long. We performed leave-one-out evaluation on these eight blocks as described in the text. In each session, the Kalman filter controlled the cursor for two blocks, and a discretized HMM on latent angle with flexible tuning likelihoods controlled the cursor for the other two blocks.

Model training data included only those timesteps for which the cursor was sufficiently far from the goal to establish a reliable movement intention vector, as in Jarosiewicz et al. (2015). We also excluded timesteps for the first second after a new target was prompted (since the ground truth, vectors from cursor to target, is less reliable there due to the participant's reaction time), and timesteps when the cursor overlaps with the goal (for which the vector from the cursor to the goal is hard to define).

---

[*]Department of Computer Science, Brown University, Providence, RI, USA.

[†]Computer Science and Artificial Intelligence Laboratory, MIT, Cambridge, MA, USA.

[‡]School of Engineering, Brown University, Providence, RI, USA; and Department of Neurology, Massachusetts General Hospital, Boston, MA, USA.

[§]Rehabilitation R&D Service, Department of Veterans Affairs Medical Center, Providence, RI, USA; and Brown Institute for Brain Science, Brown University, Providence, RI, USA.

[¶]Department of Neurology, Harvard Medical School, Boston, MA, USA.

[∥]Department of Neuroscience, Brown University, Providence, RI, USA.

[**]Present affiliation: Dept. of Neurosurgery, Stanford University, Stanford, CA, USA.

[††]Department of Computer Science, University of California, Irvine, CA, USA.

We learned the parameters of the Kalman filter measurement matrix $H$ using ridge regression, using five-fold cross-validation on the regularization constant, and computed the empirical noise covariance for $Q$. We used the parameters given later in this writeup, in everyday use by T10, for state dynamics. We also learned the parameters of the flexible likelihoods using ridge regression, again using five-fold cross-validation on the regularization constant.

The IIR filter used for preprocessing has numerator coefficients $b = \frac{1}{5}$ and denominator $a = \left[1, -\frac{4}{5}\right]$.

## 2   Clinical research session design

We ran sessions with a clinical trial participant, T10, on two separate days. We went through the same procedure in each session to generate labeled training data and to control for speed, described here. Each session is divided into a series of blocks, chunks of time a few minutes long in which the T10 engaged with the cued targets task, described in the text.

### 2.1   Bootstrapping

In order to bootstrap the system with labeled training data, we started with the computer controlling the cursor, and gradually decreased the amount of computer assistance until the participant was controlling the cursor unassisted; after this bootstrapping, the participant was in full control of the cursor. We performed this standard procedure in order to balance two constraints. On the one hand, before we have a lot of labeled training data, cursor control is poor without computer assistance. On the other hand, because neural activity is context-dependent, we get the best possible control when the amount of computer assistance in the training data is similar to the amount of computer assistance at test (Jarosiewicz et al., 2013). That is, we get the best unassisted control when training on data from a task with unassisted control.

We started each session with an open-loop block, in which the computer controlled the cursor, to get an initial set of labeled calibration data. During the open loop block, the participant imagined moving along with the cursor.

We then proceeded to three blocks with partial computer control, or error attenuation, described in Jarosiewicz et al. (2015). Suppose that at timestep $t$ the output of the decoder is $v$, which can be broken down into the component perpendicular to line from the cursor to the target $v_\perp$ and the component parallel to the target, $v_\parallel$. With the error attenuation postprocessing, if the amount of error attenuation was set to $f$, then the cursor was moved along the vector $(1 - f)v_\perp + v_\parallel$. We ran three 3-minute blocks with decreasing amounts of error attenuation: 0.8, 0.5, and 0.2. After each error attenuation block, we relearned the parameters of the flexible likelihoods with the newly collected labeled training data. After this point in the session, we did not use error attenuation, and the participant was in full control of the cursor.

### 2.2   Calibration

We then ran two 4-minute blocks, during which the cursor was controlled with MSSM. We used these 8 minutes of data to learn the parameters for both the Kalman filter and our method, and used these parameters for the remainder of the session. We used 60 spike power channels, selected using the feature selection method given in Malik et al. (2015), and we used the same 60 channels for both decoders for the rest of the session.

### 2.3   Controlling for speed

Next, we chose the speed gain for each decoder (Kalman and MSSM) in order for both decoders to have the same trimmed mean speed. To collect data needed to match speeds, we ran two 2.5 minute speed reference blocks: one controlled by the Kalman filter and the other controlled by MSSM.

We chose a desired trimmed mean speed for both decoders, and linearly extrapolated the speed gain for a given decoder such that the middle 60% of the speed distribution of that decoder's speed reference block is equal to this desired trimmed mean speed when the speed gain was retroactively applied.

After matching speeds we proceeded to the six 4-minute blocks of the ABABAB comparison, described in the text of the paper.

## 3 Task Setup

Throughout each session, we used a random target task, in which the participant was prompted to move the cursor to a goal, shown as a red circle. If the cursor overlapped the target for 0.3 seconds, the target was acquired and a new target was shown. If the participant failed to acquire the target in 10 seconds, then that acquisition timed out and new target was shown.

We designed the task so that the angle from the cursor's starting position to the goal was as uniformly distributed as possible, in order to improve our ability to learn the flexible tuning curves from this task. All goal targets were chosen within a square, centered on the center of the screen, with a side length of 0.8 distance units. When it was time to select a new target location, we followed this procedure:

- We sampled an angle $a$ uniformly from the unit circle
- Let $d$ be the length of the ray from the current cursor position to the edge of the square of allowable target locations. If $d$ is less then 0.5 distance units, then go back to the first step of this procedure.
- Choose a random distance $d'$ in the interval $(0.5, d)$.
- The new goal location is $(x + d' \cos(a), y + d' \sin(a))$, where $(x, y)$ is the current cursor position.

We approximately sampled random quantities using a pseudorandom number generator.

## 4 Hyperparameters

### 4.1 Linear Gaussian State Space Model Parameters

For the baseline Kalman filter we chose the parameterization in place for regular clinical trials of the participant at the time:

$$A = 0.9929 * I, \qquad W = 0.24 * I \tag{1}$$

where $A$ is the state transition model and $W$ the process noise covariance. Both matrices are $2 \times 2$ with the first dimension being horizontal velocity and the second dimension being vertical velocity.

### 4.2 MSSM Decoder

We set the von Mises basis function spacing, for the flexible likelihood model, equal to 22.5 degrees and $p(c)$ as uniformly distributed over 25 timesteps. We discretized the goal position $g_t$ as a $40 \times 40$ grid. Angular aim $\theta_t$ takes one of 72 discrete values on a regular grid in $[0, 2\pi)$ (that is, one discrete angle state every five degrees). Note that the cursor position is not restricted to this discrete grid as it is a function of the posterior on goal position described in the main text. Additional hyperparameters are set at:

$$p(b) = \begin{bmatrix} 0 & 0 & 0 & 0 & \frac{1}{6} & \frac{1}{6} & \frac{1}{6} & \frac{1}{6} & \frac{1}{6} & \frac{1}{6} \end{bmatrix} \tag{2}$$

$$\eta = 0.16, \qquad \alpha = \frac{1}{20}, \qquad \bar{\kappa} = 8, \qquad \epsilon = 0.4 \tag{3}$$

where $\eta$ controls the probability of sampling a new goal position, $\alpha$ is the smoothing parameter in the smooth von Mises distribution over cursor angle with concentration $\bar{\kappa}$, and $\epsilon$ is the flexible tuning bandwidth. Finally, $p(b)$ is the discrete distribution of new cursor aim counter values.

## 5 MSSM Inference

As noted in the main text we perform approximate junction tree inference using the Boyen-Koller approximation Boyen and Koller (1998). Specifically, given neural measurements $z_{1...t} = \{z_1, \ldots, z_t\}$

Figure 1: Junction tree used for realtime inference

the posterior distribution is approximated as,

$$p(g_t, c_t, \theta_t, b_t \mid z_{1...t}) \approx p(g_t, c_t \mid z_{1...t}) p(\theta_t, b_t \mid z_{1...t}). \tag{4}$$

We compute the marginals $\{g_t, c_t\}$ and $\{\theta_t, b_t\}$ using the Shafer-Shenoy algorithm (Shafer and Shenoy, 1990). Recall that $g_t$ and $c_t$ are the latent goal position and counter, respectively, while $\theta_t$ and $b_t$ are the cursor aim and corresponding counter. All quantities are discrete. To avoid exponential complexity our approximation estimates a product posterior over these quantities.

We define the auxiliary variable $r_t \triangleq a(g_t, p_t)$ as the angle from cursor to target. Here, $a(g, p) = \tan^{-1}((g_y - p_y)/(g_x - p_x))$, $p_x$ is the horizontal component of $p$, and $p_y$ is the vertical component of $p$. Because $r_t$ takes one of 72 values in in $[0, 2\pi)$, using this auxiliary variable reduces the computation of von Mises densities during inference. Note that the posterior mean on $r_t$ and $\theta_t$ can be any continuous angle. The posterior distribution decomposes into four factors, grouped below as distinct colors:

$$p(g_t, b_t, \theta_t, c_t, r_t | z_{1...t}) \propto p(g_t, c_t | z_{1...t-1}) p(r_t | g_t, p_t) p(\theta_t | r_t, z_{1...t-1}) p(z_t | \theta_t) p(b_t | \theta_t, z_{1...t-1})$$

Each factor forms a clique of the junction tree shown in Figure 1. Each clique factor is represented as a two-dimensional matrix. Some of the factors in these cliques are derived from the posterior at the previous timestep. The first factor defines the joint distribution of the goal position $g_t$ and the goal counter $c_t$ as:

$$p(g_t = g, c_t = c | z_{1...t-1}) = p(g_{t-1} = g, c_{t-1} = c + 1 | z_{1...t-1}) +$$
$$\eta \frac{1}{G} p(c) p(c_{t-1} = 0 | z_{1...t-1}) + (1 - \eta) p(c) p(g_{t-1} = g, c_{t-1} = 0 | z_{1...t-1})$$

The second factor is a delta function which enforces our auxiliary variable definition $p(r_t \mid g_t, p_t) = \mathbb{I}[r_t = a(g_t, p_t)]$. The third factor is a conditional distribution on cursor aim,

$$p(\theta_t = a | r_t, z_{1...t-1}) = \ldots$$
$$p(\theta_t = a | b_{t-1} = 0, r_t) p(b_{t-1} = 0 | z_{1...t-1}) + p(\theta_{t-1} = a, b_{t-1} \neq 0 | z_{1...t-1}).$$

where $p(\theta_t | b_{t-1} = 0, r_t) = \text{vMS}(\alpha, r_t, \bar{\kappa})$ when the aim counter is exhausted. The likelihood $p(z_t \mid \theta_t)$ is our flexible tuning model described in the main text. Finally, the conditional distribution for the aim counter is:

$$p(b_t = b | \theta_t, z_{1...t-1}) = \ldots$$
$$= p(b_t = b, b_{t-1} \neq 0 | \theta_t, z_{1...,t-1}) + p(b_t = b | \theta_t, b_{t-1} = 0) p(b_{t-1} = 0 | z_{1...t-1})$$
$$= p(b) p(b_{t-1} = 0 | z_{1...t-1}) + \frac{p(b_{t-1} = b + 1, \theta_t | z_{1...t-1})}{\sum_{b_{t-1}} p(b_{t-1}, \theta_t | z_{1...t-1})}$$

Each factor at time $t$ is represented as a two-dimensional matrix. We then perform discrete inference using the Shafer-Shenoy algorithm (Shafer and Shenoy, 1990).

# 6 Author ORCIDs

Leigh Hochberg, [ID] http://orcid.org/0000-0003-0261-2273

Daniel Milstein, [ID] https://orcid.org/0000-0003-2961-5490