[Reviews · NeurIPS 2017]

Reviewer 1



This paper presents a dynamic Bayesian network to improve the error made by iBCI decoders. This method allows the system to use longer signals history and reduce errors. It seems that the state of the art is the use of Kalman filters to decode the neural signals to find the angle of motion. If that is the case, the authors show improvement upon this method by using a better tuning curve and a modified Kalman Filter. It would be interesting to know the minimum recording time necessary to initialize the tuning curve and the decoder. The authors state: "While the Kalman filter state solely encodes the vector of desired cursor movement, we expand our decoder’s state to also include the two-dimensional final location (or goal position) on the screen to which the participant wants to move the cursor" Does this mean that the goal needs to be known beforehand? Is this a supervised approach? In that case, this limitation should be clearly pointed out.

Reviewer 2



The paper describes a novel brain-computer-interface algorithm for controlling movement of a cursor to random locations on a screen using neuronal activity (power in the "spike-spectrum" of intra-cortically implanted selected electrodes). The algorithm uses a dynamic Bayesian network model that encodes possible target location (from a set of possible positions on a 40x40 grid, layed out on the screed). Target changes can only occur once a countdown timer reaches zero (time intervals are drawn at random) at which time the target has a chance of switching location. This forces slow target location dynamics. Observations (power in spike spectrum) are assumed to be drawn from a multi modal distribution (mixture of von Mises functions) as multiple neurons may affect the power recording on a single electrode and are dependent on the current movement direction. The position is simply the integration over time of the movement direction variable (with a bit of decay). The novel method is compared with several variations of a Kalman filter (using same movement state space but missing the latent target location) and an HMM. Experiments are performed off line (using data from closed loop BCI session using either a Kalman filter or an HMM) and also in 2 online closed-loop sessions (alternating between Kalman filter and the new method). Experiments performed an a tetraplegic human subject. Overall the paper (sans supplementary material with is essential to understanding the experiment) is well written. The new model vastly outperforms the standard Kalman Filter, as is clear from the multiple figures and supplementary movies (!). Some comments: - I gather that the von Mieses distribution was used both for modelling the observations given the intended movement direction and for modelling the next goal position. I found the mathematical description of the selection of the goal position unclear. Is there a distribution over possible goal positions or is a single goal position selected at each time point during inference? Can you clarify? - Once a timer is set, can the goal be changed before the timer expires? My understanding is that it cannot, no matter what the observations show. I find this to be a strange constraint. There should be more straightforward ways of inducing slow switch rates that would not force the estimate of the target position to stay fixed as new evidence accumulates. e.g. forced transition rates or state cascades. - My understanding is that the model has no momentum. I.e., the cursor can "turn on a dime". most objects we control can not (e.g hand movements). Would it not make more sense to use a model that preserves some of the velocity and controls some of the accelleration? It is known that some neurons in motor cortex are direction sensitive and other are force sensitive. Would it not be more natural? - After an initial model learning period, the model is fixed for the rest of the session. It would be interesting to see how the level of performance changes as time passes. Is the level of performance maintained? Is there some degredation as the network / recordings drift? Is there some improvement as the subject masters control of the system? Would an adaptive system perform better? These questions are not addressed. - The paper is geared at an engineering solution to the BCI problem. There is no attempt at providing insights into what the brain encodes most naturally or how the subject (and neural system) adapt / learn to make the most out of the provided BCI. - citations are in the wrong format should be numerical [1] , [1-4] etc... not Author et al (2015).

Reviewer 3



Summary: The authors proposed a new method for cursor control in the intracortical brain-computer interfaces. To improve the performance of standard Kaman filter based decoder, they proposed a dynamic Bayesian network that includes the goal position as a latent variable, and also proposed a flexible likelihood model for neural decoding. With the proposed likelihood model and Bayesian network, they used semi-Markov model to analyze the dynamics. With a latent counter variable for both the goal position and cursor angle, their method is capable to integrate information over a longer history. They used junction tree algorithm to obtain the inference. Simulations for both offline and online experiments are used to show that their method can predict motion directions with neural activity, and has improved performance compared with Kaman filter baselines. Qualitative assessment: The paper shows that the proposed method leads to slightly improved performance compared with traditional Kaman filter decoder in the brain-computer interface system. According to the offline experiment result in Figure 5, the flexible tuning likelihood and the semi-Markov method both make a contribution to improving the result. The topic is not novel, but the work is a reasonable advance on previous work. The paper is well organized and generally clearly written. Comments: 1) In the dynamic Bayesian network proposed by the author, the goal position is incorporated as a latent state. The motivation is unclear. It seems that the system needs to infer the not only the intent of the movement, but also the final goal of the subject that uses the brain-computer interfaces, which is radically harder problem. This should be discussed more thoroughly. 2) In the offline experiment, the advantage of the proposed method can be observed compared to kaman filter decoder with various configuration. However, in the online experiment, the improvement is not obvious to me. In the left subfigure of figure 7, I can hardly conclude which gives better performance. 3) Check for typos in the derivation of the inference algorithm in the supplementary materials.